# Mitochondrial Permeability Transition, Cell Death and Neurodegeneration

**DOI:** 10.3390/cells13070648

**Published:** 2024-04-08

**Authors:** Artyom Y. Baev, Andrey Y. Vinokurov, Elena V. Potapova, Andrey V. Dunaev, Plamena R. Angelova, Andrey Y. Abramov

**Affiliations:** 1Laboratory of Experimental Biophysics, Centre for Advanced Technologies, Tashkent 100174, Uzbekistan; baev.a.yu@gmail.com; 2Department of Biophysics, Faculty of Biology, National University of Uzbekistan, Tashkent 100174, Uzbekistan; 3Cell Physiology and Pathology Laboratory, Orel State University, Orel 302026, Russia; tolmach_88@mail.ru (A.Y.V.); e.potapova@oreluniver.ru (E.V.P.); dunaev@bmecenter.ru (A.V.D.); 4Department of Clinical and Movement Neurosciences, UCL Queen Square Institute of Neurology, Queen Square, London WC1N 3BG, UK; p.stroh@ucl.ac.uk

**Keywords:** mitochondrial permeability transition, cell death, neurodegeneration, neuron, astrocyte

## Abstract

Neurodegenerative diseases are chronic conditions occurring when neurons die in specific brain regions that lead to loss of movement or cognitive functions. Despite the progress in understanding the mechanisms of this pathology, currently no cure exists to treat these types of diseases: for some of them the only help is alleviating the associated symptoms. Mitochondrial dysfunction has been shown to be involved in the pathogenesis of most the neurodegenerative disorders. The fast and transient permeability of mitochondria (the mitochondrial permeability transition, mPT) has been shown to be an initial step in the mechanism of apoptotic and necrotic cell death, which acts as a regulator of tissue regeneration for postmitotic neurons as it leads to the irreparable loss of cells and cell function. In this study, we review the role of the mitochondrial permeability transition in neuronal death in major neurodegenerative diseases, covering the inductors of mPTP opening in neurons, including the major ones—free radicals and calcium—and we discuss perspectives and difficulties in the development of a neuroprotective strategy based on the inhibition of mPTP in neurodegenerative disorders.

## 1. Introduction

One of the biggest unsolved problems in modern medicine are neurodegenerative diseases, which, due to the aging of the population in most countries worldwide, have become a serious challenge for society. Two of the most common neurodegenerative disorders—Alzheimer’s disease and Parkinson’s disease—were described for the first time more than 100 and 200 years ago, respectively, and are still incurable [1,2]. However, significant progress in the understanding of the molecular and cellular mechanisms of neurodegeneration in the last few decades has been prompted by the discovery of the toxins which could lead to specific neuronal loss as well as the discovery of the genes causing familial forms of these diseases. This enables the generation of toxic and transgenic cellular and animal models of neurodegenerative diseases which aid not only in the study of the pathogenesis of the diseases but also serve as test platforms for potential therapeutic strategies. Although the clinical manifestation and molecular mechanisms of the pathology in each neurodegenerative disease are different, all of them include mitochondrial dysfunction and the involvement of protein misfolding in their etiopathology [3].

Mitochondrial dysfunction is strongly associated with Parkinson’s disease—mitochondrial toxins have been found to induce Parkinson’s disease in animals; moreover, proteins which are encoded by genes associated with Parkinson’s disease are mitochondrial or mitochondria-associated [3,4]. Various forms of mitochondrial dysfunction have also been found in Alzheimer’s disease, ALS, frontotemporal dementia and other neurological conditions [5,6,7,8]. However, the term “mitochondrial dysfunction” may include changes in a variety of functions in the mitochondria, from the alteration of the mitochondrial electron transport chain (ETC) to the overproduction of reactive oxygen species (ROS) [9] and defects in the mitochondrial Ca^2+^ transport [10,11]. The combination of cellular and mitochondrial pathological events triggers the mechanism of cell death which initiates with mitochondrial swelling and the release of pro-apoptotic proteins [12]. Considering this, neuronal cell death and neurodegeneration starts from a mitochondrial event known as the mitochondrial permeability transition, a non-selective permeability of the mitochondrial membrane which leads to the swelling of the mitochondria, followed by protein release (including cytochrome c) and subsequent cell death.

Despite much evidence for this pathway, the direct link between mPTP and neurodegeneration is still debatable for several reasons: (a) the nature of the mitochondrial permeability transition and the molecular structure of the pore are still controversial; (b) the detection of mPT in live cells or tissues is methodologically very difficult; (c) the threshold of the mPTP opening that triggers cell death, considering the possible role of mPTP in the physiology, is still under investigation.

The inability of neurons from the central nervous system to regenerate makes the clarification of the mechanism of apoptosis initiation vitally important for the development of neuroprotective strategies. Here, we review the possible structure of mPTP, the mechanism of its activation and discuss how this mitochondrial event leads to neuronal loss in the major neurodegenerative diseases.

## 2. Permeability Transition

The non-selective permeability of mitochondrial membranes (permeability transition, PT) is characterized by an increase in the permeability of mitochondrial membranes for various ions and substances whose molecular weight does not exceed 1500 Daltons (Da). This phenomenon is accompanied by the loss of membrane potential, the dissipation of the proton gradient across the inner membrane and mitochondrial swelling. Uncontrolled swelling leads to the rupture of the mitochondrial outer membrane. The phenomenon of mitochondrial swelling was discovered and actively studied in the 1950s and 1960s. Using isolated mitochondria, it was found that Ca^2+^, fatty acids, inorganic phosphate and substances that induce oxidative stress contribute to mitochondrial swelling, while Mg^2+^ and adenine nucleotides suppress this process [13,14,15,16,17,18]. In 1979, Haworth and Hunter characterized the main indicators of Ca^2+^-induced membrane transition and named the potential structure that triggers the mitochondrial PT a PT pore—mPTP [19,20,21]. However, the discovery of the mPTP inhibitor cyclosporine A (CsA) [22] helped to link this exclusively mitochondrial phenomenon to the physiology and pathology of the cells. CsA is a cyclic non-ribosomal polypeptide consisting of 11 amino acids produced by the soil fungus *Beauveria nivea*. CsA is widely used as an immunosuppressant, inhibiting peptidyl-prolyl cis–trans isomerases [23,24] as well as its mitochondrial form, cyclophilin D (CypD) [23]. Over the 70-year history of mPTP research, numerous activators and inhibitors of this pore have been discovered but CsA is still the most commonly used inhibitor of mPTP [25]. The size of the pore was identified using solutions of molecules with different sizes, while the electrophysiological assessment, based only on the inner membrane currents (using mitoplasts), helped to identify the current flowing through the pore [26]. For many researchers, the term mPTP is directly associated with pathology and cell death; however, transient mPTP opening (flickering) is an essential process for maintaining mitochondrial physiology, and is involved in the efflux of matrix Ca^2+^ and ROS [27].

For several decades, the molecular nature of mPTP was unknown and scientists tried to identify the key pore-forming constituents of mPTP (Figure 1). The first candidates for this role were the ATP/ADP transporter (ANT) as a pore-forming protein in the inner membrane and the voltage-dependent anion channel (VDAC) as a pore-forming protein in the outer membrane. In this model, the translocator protein, hexokinase 2 (HK2), the mitochondrial creatine kinase (MtCK) and CypD were assigned a stabilizing and regulatory role (Figure 1A). A hypothesis about ANT involvement in the PTP formation was made due to the discovery that ANT inhibitors such as atractyloside and bongkrekic acid affect the PTP activity. Moreover, the molecular complex of ANT with VDAC, HK2, MtCK and CypD, isolated from the mitochondria and reconstituted into liposomes, have shown biophysical properties resembling those of PTP [28,29,30]. However, in one study, when genetically inactivated, the two isoforms of ANT could not inhibit calcium-dependent mPT [31]. The observed mPT may have been inhibited by CsA and activated by thiol oxidants. However, in 2019 it was shown that the genetic deletion of the three isoforms of ANT (Ant1, Ant2 and Ant4) and CypD abolished Ca^2+^-induced mPTP formation in mice mitochondria [32], which brought ANT back on the radar of scientists as a potential pore-forming component.

From the very beginning until now CypD has been the only protein whose forming role in the PTP phenomenon has been 100% proven. In this regard, when new experimental data shows the interaction of CypD with any mitochondrial inner membrane proteins, this protein immediately begins to be considered as a potential mPTP-forming protein. In 2008 it was found that CypD can directly interact with the phosphate carrier (PiC) and ANT; thus, PiC became the new PTP-forming candidate (Figure 1B) [33]. However, an earlier electrophysiological evaluation of PiC purified and reconstituted into giant liposomes showed that it is able to generate Ca^2+^-induced currents, with a mean conductance ranging from 25 to 40 pS in different experimental conditions, which is much lower compared to PTP currents previously observed [34]. Recent work, using genetic ablation, has shown that PiC is not an obligatory part of PTP, but certainly participates in its regulation [35]. Moreover, phosphate is one of the major regulators of the mPTP opening along with the endogenous inorganic polyphosphate (polyP). Thus, the reduction of the mitochondrial polyP by the overexpression of mitochondrially targeted polyphosphatase has been shown to inhibit Ca^2+^-induced but not ROS-induced PTP and to protect mammalian neurons and astrocytes from Ca^2+^-induced and beta-amyloid-induced cell death [36], which was also confirmed in isolated rat liver mitochondria [37]. It should be noted that long-chain polyPs induce mitochondrial depolarization and apoptotic cell death, which can be prevented by CsA [38].

One of the most recent and promising findings is that the mitochondrial F_1_F_O_-ATP synthase could potentially form the mPTP [39,40] (Figure 1C,D). However, despite the number of studies directly or indirectly supporting the hypothesis of PTP formation by the ATP synthase, it remains unclear how this complex enzyme switches from ATP synthase/ase into mPTP.

It should be noted that the potential mPT players are not restricted to mitochondrial proteins. Thus, palmitic acid not only induces CsA-dependent mPT but also initiates apoptosis [41,42].

However, despite the multiple candidates for molecular contributors to the mPTP complex or for serving as inductors or inhibitors of the pore opening, the evidence implicating mPTP in cellular and whole animal pathology remains unequivocally contingent upon Ca^2+^ or redox dynamics, which in turn are dependent on CsA and CypD. However, despite the fact that CsA is a very efficient CypD inhibitor, it is not used in clinics as an mPTP inhibitor. CsA is an FDA-approved immunosuppressive agent, which might be used during solid organ transplantations, for rheumatoid arthritis, psoriasis, ALS, nephrotic syndrome, graft vs. host disease, etc. Moreover, CsA is quite toxic, and its application has a lot of side effects, so research and development of new non-immunosuppressive CypD and mPTP inhibitors remains an important and challenging task. Therefore, in the pursuit of discovering novel inhibitors targeting CypD and mPTP, researchers have used various approaches, ranging from the large-scale screening of substances available in FDA-approved and other libraries [43,44] to the de novo development of new peptides [45]. Some of the new mPTP inhibitors (Ebselen) have already proved their efficiency in mouse models of Alzheimer’s disease [44].

## 3. Alzheimer’s Disease

The most common neurodegenerative disease, Alzheimer’s disease, is also the most common cause of dementia. Major histopathological features of this disease include extracellular senile plaques (consisting of the aggregated β-amyloid) and intracellular neurofibrillary tangles (formed by aggregated tau protein). Familial forms of the disease are caused by mutations in the amyloid precursor’s proteins or presenilins, leading to the misfolding and aggregation of β-amyloid; importantly, the aggregated β-amyloid is neurotoxic and for decades has been used for the induction of a cellular model of Alzheimer’s disease [46,47].

β-amyloid in aggregated form has been shown to be membrane-active and forms calcium channels in artificial membranes and in the membranes of astrocytes [48,49] as well as directly induces the opening of mPTP in experiments with isolated brain mitochondria [50,51]. In astrocytes, the combination of a β-amyloid-induced calcium signal and ROS overproduction has led to mitochondrial depolarization, which was dependent on CsA, inhibitors of mitochondrial calcium uptake, or it was blocked in astrocytes from CypD knockout mice [5,52,53].

It should be noted that the triggering of cell death through mPTP has also been demonstrated in the familial forms of Alzheimer’s disease. Thus, presenilin-1 mutation is linked to perturbated calcium homeostasis and oxyradical production, which trigger mPTP and induce caspases activation [54]; in addition, both presenilin-1- and presenilin-2-associated mutations increase the gain of function of IP_3_ receptors and the open probability of mPTP [55] (Figure 2).

The role of mPTP in neuronal loss through Alzheimer’s disease is also shown in experiments with animal models of this disorder. Thus, CypD deficiency protects neuronal function and ameliorates learning and memory deficits in β-amyloid and APP-mutated mice [56,57]. Interestingly, CypD deficiency protects the function of F_O_F_1_-ATP synthase, one of the candidates for constituents of mPTP [58].

Importantly, the major trigger for mPTP—mitochondrial calcium overload—is connected to the inhibition of the mitochondrial calcium efflux in the pathology [10,59,60]. Importantly, the molecular inhibition of the mitochondrial calcium uptake mitigates the pathology in neurodegeneration [61], and the pharmacological sequestration of the mitochondrial calcium uptake protects against excitotoxicity [62] and dementia in Alzheimer’s disease models (Figure 2).

The induction of mPTP in Alzheimer’s disease requires several factors. Thus, despite the mitochondrial calcium overload, the inhibition of reactive oxygen species in NADPH oxidase, upon the application of β-amyloid, completely blocks mitochondrial depolarization [5]. However, the direct delivery of singlet oxygen to the mitochondria in live cells without mitochondrial calcium overload may not induce the opening of mPTP and cell death [53,63] (Figure 2).

Based on the importance of mPTP in the pathology of Alzheimer’s disease, a number of studies have focused on the study of novel molecules for the inhibition of the permeability transition and neuroprotection [44,64].

## 4. Parkinson’s Disease

Parkinson’s disease is the most common movement disorder and the second most common neurodegenerative disease. It is characterized by the loss of dopaminergic neurons in *substantia nigra*, as are all other neurodegenerative disorders associated with the deposition of misfolded proteins (i.e., aggregated α-synuclein) as intracellular inclusions, or Lewy bodies. More than any other neurodegenerative disorder, it is associated with mitochondria, due to the induction in Parkinson’s disease of mitochondrial toxins and mitochondrial localization, or the connection to proteins encoded by the Parkinson’s genes [3]. However, not all mitochondrial toxins or mutations in complex I lead to Parkinson’s disease [65,66].

The physiological function of monomeric α-synuclein is connected to signal transduction but also to the function of the mitochondrial F_O_F_1_-ATP synthase. Binding to this enzyme, monomeric α-synuclein increases the efficiency of ATP synthesis [67]. Both monomeric and oligomeric (toxic) α-synucleins are able to incorporate themselves into the plasma membrane and induce a calcium signal [68,69]. However, only oligomeric α-synuclein (not fibrils or monomers) produces reactive oxygen species [69,70]. The oligomeric α-synuclein form, which can produce ROS, binds to the F_O_-F_1_-ATP synthase in the same way as physiological α-synuclein monomers and oxidases, an enzyme which triggers mPTP opening and cell death [71], and which also confirms the possible involvement of complex V in the mPTP structure (Figure 3).

The toxicity of the inhibitors of complex I which induce Parkinson’s disease can be partially blocked by CsA or CypD deficiency, confirming the involvement of permeability transition in neurodegeneration [72,73,74].

Familial forms of Parkinson’s disease—PINK1 and LRRK2 mutations—have slower mitochondrial calcium efflux due to the inhibition of the enzyme NCLX [75,76,77] that renders these cells vulnerable to dopamine due to calcium overload and ROS production, leading to mPTP opening [78] (Figure 3).

The familial forms of Parkinson’s disease show no strong phenotype in rodents but induce Parkinson’s-like movements in Drosophila. CypD mutation ameliorates oxidative stress-induced defects in a Drosophila DJ-1 null mutant that links the phenotype and neuronal loss to the mitochondrial permeability transition [79].

## 5. Amyotrophic Lateral Sclerosis

Amyotrophic lateral sclerosis (ALS) is associated with damage to upper and lower motor neurons in the cerebral cortex and spinal cord, as well as skeletal muscle dystrophy [80]. The prevalence of ALS, according to different data, ranges from 0.6 to 6 cases per 100,000 people, with a significant difference between ethnic groups [81]. The average age of diagnosis is 55–65 years, but familial forms are characterized by an earlier onset (47–52 years). The average life expectancy of patients is 2–3 years from the time of diagnosis. The rapidly developing symptomatology of the disease includes weakness, twitching, spasticity and respiratory failure. About 5–10% of registered cases of ALS are related to familial forms [82,83].

Mitochondrial dysfunction, including that which is associated with mPTP opening, is considered to be one of the key causes of death in both motor neurons and muscle cells in ALS [84,85]. Such ALS-related pathological processes as excitotoxicity [86] with a reduced calcium buffering capacity of the cytosol and mitochondria [87], overproduction of ROS and ONOO- [88], nitration of proteins including CypD and ANT [89], ATP level decrease, disturbances in mitochondrial dynamics [90], mitochondrial swelling and vacuolization [91], appear as triggers and consequences of mPTP opening [92]. Mouse models of ALS also show increased expression of mPTP-associated proteins (VDAC, ANT, CypD) with increased accumulation in mitochondria [89]. Moreover, these changes are often found in both motor neurons and muscle cells [93]. The neurotoxicity of the mutant protein TDP-43 penetrating into mitochondria may be related to the mPTP opening, mtDNA release and the activation of inflammation via the cGAS/STING pathway caused by increased mtROS production [94].

The role of mPTP is also confirmed by the change in pathology progression when modulating individual proteins influencing the pore. For example, mice with SOD1 mutations and lacking the Ppif gene encoding CypD showed improved mitochondrial function and morphology, decreased accumulation of the mutant protein and motor neuron death [87]. In the SOD1G93A model, the increased expression of Bcl-2 inhibiting mPTP opening resulted in a delayed onset of the pathology, as well as an increased lifespan of transgenic animals [95].

mPTP plays a dual role relative to cell life and death, and to date there is no definite explanation of the mPTP opening effect on the viability of motor neurons and muscle cells; moreover, some data are controversial. For a number of models, the classical pathway of apoptosis has been shown to increase the activity of caspase-1 and caspase-3 with a concomitant increase in CypD accumulation in defective mitochondria [96]. In addition, the ability to bind the anti-apoptotic protein Bcl-2 [97] and to increase the expression of pro-apoptotic proteins Bim and Bax [98] was revealed for mutant SOD1. A decreased rate of symptom development and an increased lifespan of SOD1-transgenic mice was obtained by the genetic knockout of Bim, Bax and Bak, and by the inhibition of caspases [99,100]. However, in cell models with the mutant TDP-43 protein, a relatively small increase in apoptotic cells was observed [101]. It has also been shown that despite the development of chromatin condensation, the morphology of dead cells differs from the classical pattern of apoptosis [91]. The changes in the mitochondrial state occurring in ALS, which may be associated with mPTP opening, in some cases do not lead to the development of apoptosis [88]; in particular, they do not activate caspase-1 and caspase-3 [91].

The efficacy of the ALS treatment strategies used in modern medical practice is extremely low. Riluzole [102], which reduces the effect of excitotoxicity, as well as the radical scavenger Edaravone [103], allow for delaying the progression of disease symptoms for several months. Therefore, the shown association of mPTP with ALS makes the pore an attractive therapeutic target.

Thus, the use of CypD inhibitors, in particular the administration of CsA (intraventricular or intraperitoneal with increasing permeability of the blood–brain barrier) can lead to an enhanced lifespan of SOD1G93A model animals. CsA also reduces mitochondrial depolarization in the motor neurons of SOD1G93A and SOD1G85R models under repetitive stimulation [104]. The stabilization of mitochondrial creatine kinase and the inhibition of mPTP opening may explain the improved motor function and increased survival of SOD1G93A-transgenic mice on a creatine-enriched diet [105]. The application of another mPTP inhibitor—GNX-4728—increased calcium retention capacity, decreased mitochondrial swelling and the motor neuron death rate and resulted in an almost two-fold increase in the lifespan of animals carrying the mutant G37R allele of the SOD1 gene [106]. The application of the mitochondrial-targeted neuroprotector Olesoxime (TRO19622) has also shown promising results with ALS cellular and animal models. The application of Olesoxime increased motoneuron viability, reduced glial cell degeneration, delayed the onset of the pathology phenotype and increased the survival of the animals [107]. Many authors suggest that the positive effect of Olesoxime on mPTP is related to its ability to bind to TSPO and VDAC [107]; however, previous research has shown that these proteins do not play any significant role in PTP [108,109]. Considering the nature of the motoneurons, mPTP opening is the initial trigger for neuronal loss in ALS.

## 6. Frontotemporal Dementia

Frontotemporal dementia (FTD) has a number of phenotypic manifestations (the most common are behavioral (bvFTD) and language variants (lvFTD)) [110,111] and is characterized by the selective degeneration of the frontal and/or temporal lobe [112]. FTD is the second cause of degenerative dementia after Alzheimer’s disease (AD), distinguished by an earlier onset with diagnosis between the ages of 45–65 years [113]. With a prevalence of 15–22 cases per 100,000 people [114], familial forms account for 30–50% of patients with FTD [112]. Often, fFTDs are associated with ALS and AD or other tauopathies-related genes [115,116]. Mutations in ABCA7, CTSF [117] and PGRN [118] are specific to FTD. The main published FTD research has been performed on cellular (iPSC-derived neurons in particular) and animal models mainly with mutations of the MAPT gene encoding tau protein.

One study on tau−/− mice, which showed improvement in the mitochondrial function and cognitive abilities of animals with a significant decrease in the level of CypD expression, indicates that there is a connection between tau and mPTP regulation and, consequently, cell viability [119]. Meanwhile, the presence of mutant tau in cells leads to pore opening conditions. These include oxidative stress, impaired calcium homeostasis, inflammation and excitotoxicity [120]. In particular, iPSC-derived neurons with a 10 + 16 MAPT mutation leading to FTD-associated protein splicing have been shown to exhibit increased ΔΨm due to complex V reversal mode, increased ROS production and finally cell death [6]. The primary neuroglial cultures treated with tau K18, as well as the iPSC-derived neurons with the 10 + 16 MAPT mutation, have shown impaired calcium homeostasis manifested in the oscillations of Ca^2+^ concentration in the cytoplasm, as well as in the decreased rate of Ca^2+^ efflux from mitochondria. As a result, the cells appear to be more sensitive to the mPTP opening-associated apoptosis stimulated by the calcium overload after physiological stimuli [60]. The significant reduction of apoptosis in the presence of MitoTempo suggests a key role for mitochondrial ROS in this process [121]. In addition to stimulating oxidative stress in cells with 10 + 16 MAPT mutation, mtROS lead to the increased expression of AMPA and NMDA receptors, which can also be reduced by mitochondrial antioxidants [122]. In a mouse model, it was shown that full-length tau with the A152T mutation causes NMDA receptor-related excitotoxicity with increased extracellular glutamate, which can be reduced by stimulating the astrocytic glutamate uptake via the EAAT2/GLT1 transporter [123]. An increase in intracellular calcium and subsequent neuronal death have also been shown for tau oligomers with the R406W mutation, which reduces the microtubule binding of tau, as well as for the 10 + 14C→T mutation, which increases the 4R/3R ratio [124].

FTD-related tau mutations, which result in altered 4R/3R ratios and reduce the protein’s ability to bind to the microtubules, increase the sensitivity of neurons to apoptosis associated with an increase in intracellular calcium [125]. Studies using selective inhibitors show a regulatory role of mutant tau with respect to L-type calcium channels [126].

The increase in caspase activity that occurs as a result of mPTP opening may itself lead to the increased neurotoxicity of tau. Neurotoxic fragments, particularly NH2-26-44, which form as a result of tau hydrolysis, bind to cytochrome c oxidase and ANT and lead to the uncoupling of oxidative phosphorylation [127].

At the same time, it should be noted that tau can also trigger alternative mechanisms of apoptosis with similarities to mPTP opening. Camilleri et al. [128] have shown that mitochondrial swelling, cytochrome c release and the loss of ΔΨm under the influence of mutant hTau46 were associated with an increase in mitochondrial membrane permeability due to the interaction of tau with cardiolipin [128]. Thus, mPTP opening marks the initial step in the neurodegeneration of FTD.

## 7. Huntington’s Disease

Huntington’s disease (HD) is an autosomal dominant neurodegenerative disease, caused by a mutation that expands a specific sequence of nucleotides (CAG repeats) within exon 1 of the HTT gene. This mutation causes the production of an abnormal protein, the mutant huntingtin (mHtt), which can lead to the development of symptoms such as chorea, cognitive decline and psychiatric issues [129]. The mHtt is able to initiate a cascade of molecular damage processes, which eventually leads to mitochondrial dysfunction, ROS formation and increased oxidative stress [130,131,132]. The mitochondria in the striatum may be considered the place where various stimuli are integrated, including calcium homeostasis [3,133].

Increased mitochondrial calcium is one of the major factors for mPTP activation. Mitochondria isolated from cells obtained from both HD patients and HD mouse models are more sensitive to Ca^2+^ overloads, i.e., they have a significantly lower threshold required to trigger mPTP [134,135,136,137,138,139]. Choo et al. have demonstrated that mHtt directly induces the mPTP in isolated mouse liver mitochondria. This effect is completely prevented by CsA and exogenous ATP. It has been suggested that mHtt may promote mPTP induction by increasing the binding affinity of the ANT to mPTP activators, such as calcium and CsA, the main regulator of mPTP induction. It also prevents the binding of mPTP inhibitors, such as adenine nucleotides [135].

An artificial chromosome (YAC) transgenic mouse model of HD has also demonstrated the relationship between calcium signaling impairment and apoptosis in medium spiny neurons (MSN) in HD. The stimulation of glutamate receptors causes abnormal calcium reactions in human dopaminergic neurons (HD MSN) and cytosolic Ca^2+^ overload. Over time, the capacity of mitochondria to store Ca^2+^ is exceeded, leading to the opening of the mPTP, the release of cytochrome c into the cell and apoptosis [136].

Studies on mitochondria from mutant (STHdhQ111/Q111) huntingtin-expressing cells of striatal origin have shown that these cells have mitochondrial Ca^2+^ handling defects and that the increased sensitivity to Ca^2+^ induced mitochondrial permeabilization. At the same time, the mitochondrial dysfunction caused by Ca^2+^ overload can be reversed using CsA [138]. The study of the muscles and muscle mitochondria of 14-to-16-week-old R6/2 mice, as a model for HD, allowed for the confirmation of the decreased stability of HD mitochondria against the Ca^2+^-induced permeability transition [139].

Similar observations were obtained in neuronal cultures, in which it was shown that mHtt reduced the content of mitochondrial Ca^2+^ calcium by increasing the susceptibility to mPTP induction [140].

Gellerich et al. conducted research on the mitochondria in the brains of transgenic HD rats with the 51-glutamine repeat (htt 51Q) and found that the mitochondrial toxicity of htt 51Q appears to be affecting the regulatory binding sites of Ca^2+^-induced mitochondrial carrier proteins, such as Aralar and mPTP. This ultimately leads to energy depletion, cell death and tissue atrophy [137].

R. Quintanilla has made a significant contribution to our understanding of the role of mPTP in mitochondrial damage caused by mHtt. When observing isolated mitochondria from cell lines obtained from mutant huntingtin knock-in (HdhQ111/Q111) mice, it was found that mitochondrial dysfunction in these cells could be triggered by cumulative cytosolic Ca^2+^ increases. These effects could contribute to the progression of the striatal neuron death that is observed in HD. At the same time, a Ca^2+^ overload was discovered in these mutant cells, which led to the discovery of the mPTP. However, this event could have been prevented by pretreating the cells with CsA [141].

The neuroprotective effects of mPTP inhibitors have also proven their role in HD. The mitochondria of HD patients have elevated levels of CypD activity; moreover, CypD activity increases throughout the progression of the disease [142]. The application of CsA—a selective inhibitor of CypD and mPTP—has protective effects in a mouse model of the disease [143]. Additionally, the application of another mPTP inhibitor—GNX-4728—significantly reduced the level of apoptosis in mouse striatum cells (STHdh) expressing endogenous levels of the wild-type huntingtin (STHdh^7/7^) or mutant (STHdh111/111) proteins [144].

However, it should be mentioned that other researchers have not been able to detect a significant contribution of mPTP to mitochondrial damage in HD [145,146,147,148]. Discrepancies in the observed data may be attributed to the particular model being investigated, experimental research protocols or the necessity to explore additional mechanisms in Huntington’s disease pathogenesis, such as alterations in mitochondrial dynamics [137,149]. However, recent studies also show that presymptomatic YAC128 mouse striatal mitochondria have an altered morphology and Ca^2+^ handling [150].

In conclusion, it is important to note that the mechanisms leading to neuronal death in HD have not been fully elucidated, and the question of whether mHtt increases the sensitivity of brain mitochondria and, in particular, the mitochondria of neurons to Ca^2+^ -induced damage, remains open.

## 8. Epilepsy, Multiple Sclerosis and mtDNA Mutations

Epilepsy is one of the common neurological diseases. Epilepsy is a chronic neurological disorder characterized by unprovoked and repeated seizures that occurs in millions of people globally [151]. Oxidative stress and mitochondrial dysfunction are frequently observed in epilepsy [152]. Several forms of epilepsy are associated with impaired mitochondrial function and increased ROS generation [153]. Recently, it was shown that seizure-induced calcium oscillations in primary neuronal co-cultures negatively affect mitochondrial bioenergetics, and the application of CsA prevented seizure-induced mitochondrial depolarization, thus showing the involvement of mPTP in current mitochondrial dysfunction [154]. Another group showed that ketone bodies, which are the main effectors of anti-seizure effects in the ketogenic diet modulate mPTP by increasing the calcium retention capacity of brain mitochondria isolated from the Kcna1-null mice [155].

In addition, a possible connection has been shown between the increase in the resistance of mitochondria to the induction of the opening of the mPTP and the decrease in the severity of the convulsive state in pilocarpine-induced status epilepticus and the stimulation of neurogenesis in the adult brain of mice, which makes it possible to use mitoprotective strategies in the search for anticonvulsants and C [156,157,158].

Thus, it is now generally accepted that mitochondria and the process of mPTP opening are a key step in the initiation of cell death and, accordingly, a promising target in the search for neuroprotective drugs. In this regard, the use of mitochondria as a target for the creation of a new generation of drugs that can have a neuroprotective effect, as well as act as cognitive and metabolic stimulants, is of particular scientific interest and can be of great practical importance.

Multiple sclerosis (MS) is a demyelinating, inflammatory-mediated and neurodegenerative disease affecting the central nervous system. It causes loss of neurons and damage to axons, leading to the atrophy of the central nervous system and permanent neurological and clinical disability. It has been proven that the key factors of axonal degeneration are mitochondrial dysfunction and the irreversible opening of the mPPT [159]. As previously mentioned, CypD is a key regulator of the mPPT. Forte et al. have shown that the genetic inactivation of CypD reduces axonal damage and improves disease severity in an experimental autoimmune encephalomyelitis (EAE) model of MS. The model mice lacking CyPD developed EAE, but unlike the WT mice, they partially recovered [160]. Su et al. suggested that the physiological trigger for the opening of mPTP in MS may be exposure to ROS. In the presence of pathological levels of ROS, the expression of the adaptor protein p66shc in the mitochondrial intermembrane space was increased [161]. There, p66ShcA oxidizes cytochrome c and reduces oxygen to form mitochondrial ROS, which induces the opening of mPTP. At the same time, the genetic inactivation of p66ShcA has a neuroprotective effect in a murine model of MS [162]. The synthesis of a new mitochondrial Cyp inhibitor, JW47, based on the quinolinium cation associated with cyclosporine, has also confirmed the possibility of protection against neurodegeneration in experimental MS by inhibiting mPTP. When tested on an experimental EAE model, it was shown that JW47 is a highly effective inhibitor of Ca^2+^-mediated mPTP formation and has neuroprotective properties and slows down the accumulation of disability [163].

The close relationship between mitochondria and neurodegenerative diseases as well as the neurological symptoms of some other pathologies requires attention to an intrinsic cause of mitochondrial dysfunction. Mitochondrial DNA (mtDNA) mutations seem to be one the most important among them. mtDNA encodes 13 ETC polypeptides (seven of complex I, one of complex III, three of complex IV, two of F_1_F_O_-ATP synthase) as well as 22 tRNAs and two rRNAs required for protein synthesis, which makes mtDNA quality extremely significant for mitochondrial functions. Located close to the respiratory chain, mtDNA is permanently damaged by ROS mostly produced by complexes I and III that, due to the inefficient repair system, leads to the time-dependent accumulation of mtDNA mutation [164]. The latter correlates with the time of onset of age-related pathologies and may be considered one of the causes of sporadic forms of neurodegenerative diseases [165]. Mutations in mtDNA are known in the case of AD [166], PD [167], HD and ALS [168], multiple system atrophy and dementia with Lewy bodies [164], MELAS, LHON, MERRF, Leigh syndrome and encephalomyopathy [169]. It should be noted that the accumulation of mtDNA mutations can be brain region-specific [170].

Many of the consequences of mtDNA mutations (decreased ETC complexes content, OXPHOS efficiency, ATP formation, disturbance in mitochondrial calcium buffering, increased ROS production and oxidative stress development, apoptosis) [171,172] can lead to the mPTP opening. Thus, in analyzing AD brains, the authors of one study speculate about a higher sensibility to mPTP opening in cells harboring mutations of the mtDNA control region [166]. The mutations m.9821Adel (in MT-TR gene), m.T6589C (MT-COX1), m.13887Cins (MT-ND6) and m.G12273A (MT-ND5) in neurons were associated with ETC complexes deficiency and a change in Ca^2+^ handling [172]. The increased frequency of mtDNA damage and subsequent increase of apoptosis was shown in an HD in vitro model with mutant huntingtin expression [173]. In the mouse model mutant huntingtin was associated with a two-fold higher level of mtDNA mutations, mitochondrial calcium overload and elevated superoxide production that can lead to mPTP opening [174]. In SOD1-transgenic mice, damaged mtDNA was involved in subsequent motor neuron death [175]. According to the research [176], the influence of mtDNA mutations can be linked to the interactions of aberrant mitochondrially encoded proteins with pro- and anti-apoptotic Bak, Bax and Bcl-2, regulating outer mitochondrial membrane permeability and cytochrome c release.

Unfortunately, most of the models of neurodegenerative diseases used do not allow us to determine whether DNA mutations were the primary cause of the consequences, or if they are the result of another trigger influence.

## 9. Conclusions and Perspectives

There are no doubts about the involvement of mPTP in the mechanism of neurodegeneration. It acts as an initial trigger for apoptotic neuronal death, and some data suggest that the opening of mPTP may be induced in the processes of necrosis or ferroptosis after the initial damage of the plasma membrane. Considering this, the inhibition of mPTP is a promising therapeutic target for the prevention of neuronal loss in neurodegenerative disorders. Multiple in vitro and in vivo studies demonstrate neuroprotection by the molecular or pharmacological inhibition of the mPTP [177]. However, cyclosporine A was used in medicine for decades in patients after transplantation surgery and this treatment did not show any significant changes in the cognitive decline of the patients or even show an overall impaired cognitive function compared with controls [178]. The deficiency of CypD also induces an alteration in mitochondrial energy metabolism. On the other hand, some compounds which induce mPTP opening and the activation of apoptosis, such as the electrogenic calcium ionophore ferutinin [179,180], are used as selective estrogen receptor modulators, without any neurotoxic effects [181].

However, the development of a neuroprotective strategy using inhibitors of mPTP has a number of challenges that are connected to the uncertainty of the structure of the pore and thus to a delay in the development of more specific pharmacological agents. The opening of mPTP is the last step before the initiation of the process of cell death, and the prevention of the conditions that lead to permeability transition (mitochondrial calcium overload, ROS overproduction) may be a more efficient strategy for neuroprotection.

## Figures and Tables

**Figure 1 cells-13-00648-f001:**
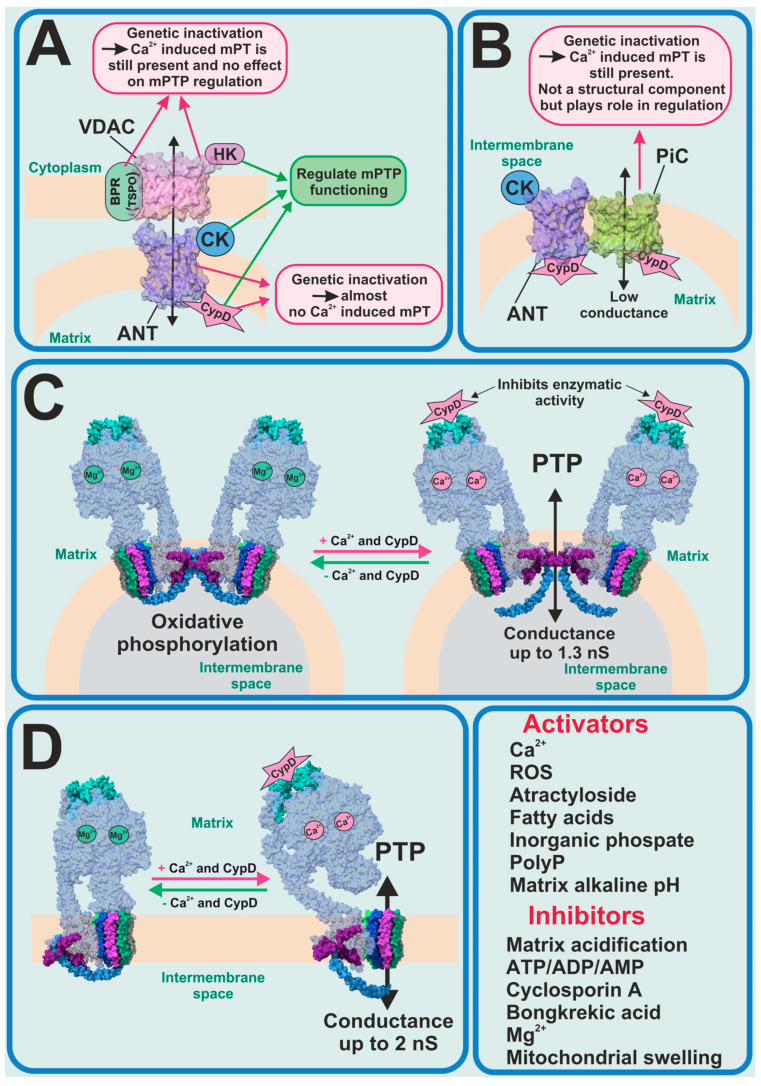
Previous and current PTP candidates. (**A**) ANT–VDAC model of PTP—partially refuted; (**B**) mitochondria phosphate carrier model of PTP—refuted; F_O_F_1_-ATP synthase as candidate for PTP-forming protein in its dimeric (**C**) and monomeric (**D**) forms.

**Figure 2 cells-13-00648-f002:**
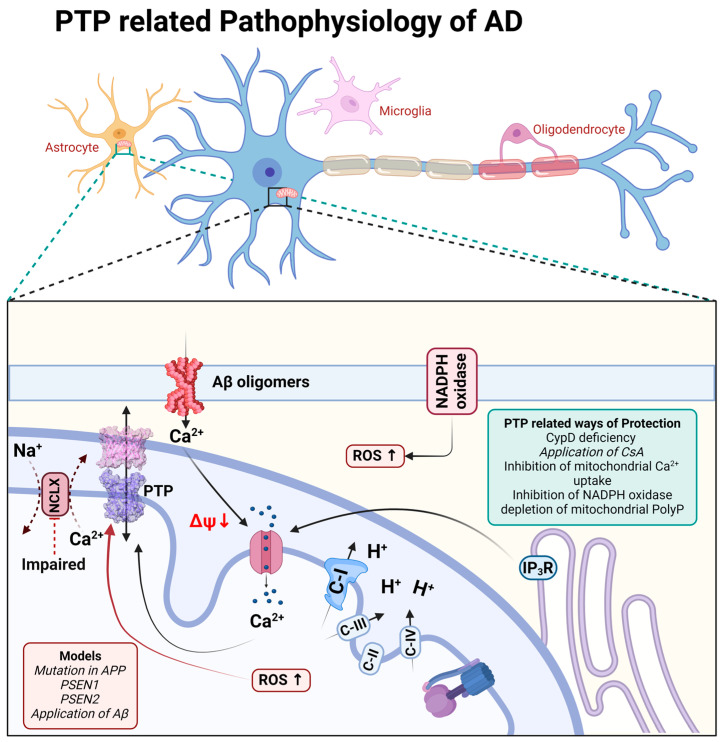
Schematic representation of PTP-related pathophysiological processes during Alzheimer’s disease.

**Figure 3 cells-13-00648-f003:**
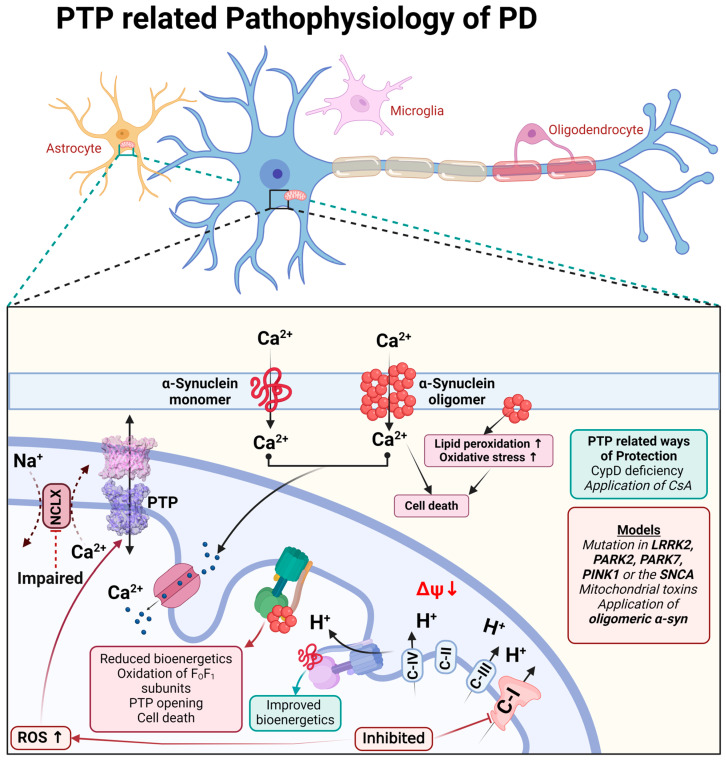
Schematic representation of PTP-related pathophysiological processes during Parkinson’s disease.

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
