# Peer review of "Mitochondrial Permeability Transition, Cell Death and Neurodegeneration"

_cells, 2024, doi:10.3390/cells13070648_

Round 1
Reviewer 1 Report
Comments and Suggestions for Authors
This review focuses on the involvement of mitochondrial Permeability Transition Pore (mPTP) in the processes of cell death in various forms of neurodegeneration. The review is well structured, the relationship of different forms of neurodegeneration with the activation of mPTP is evaluated in separate sections. The refereeing author of the article is a recognized expert in the field of neurodegeneration, with specialized expertise in mitochondrial dysfunction across diverse manifestations of neurodegenerative conditions. The text of the article is supported by 3 figures/schemes that display the potential components of mPTP, as well as the pathophysiology of Alzheimer's and Parkinson's disease connected with the activation of mPTP. The schemes perfectly convey modern ideas about the processes of neurodegeneration associated with the activation of mPTP.
I am confident that this review will pique the interest of the readership of the journal.
Minor comment:
In Fig. 1A-D it is necessary to indicate the mitochondrial matrix and intermembrane space.
Author Response
We thank the reviewer for finding our manuscript to be interesting.
We have modified Figure 1 A-D according to the comment.
Reviewer 2 Report
Comments and Suggestions for Authors
Two important areas of research have not been sufficiently addressed.
The first is the physiological regulation of mPTP by mitochondrial membrane potential, redox state, pH, etc. The role of mitochondrial redox state in neurodegeneration has been partially covered in a previous review by the authors (ref 11 ), but has not been discussed in relation to regulation mPTP.
Second, studies of new non-immunosuppressive mPTP inhibitors. New non-peptide CypD inhibitors have recently been discovered using various screening approaches (Peterson AA, Rangwala AM, Thakur MK, et al. Discovery and molecular basis of subtype-selective cyclophilin inhibitors. Nat Chem Biol. 2022; 18(11): 1184 -1195 doi:10.1038/s41589-022-01116-1; Van Bael J, Vandenbulcke A, Ahmed-Belkacem A, et al. Small-Molecule Cyclophilin Inhibitors Potently Reduce Platelet Procoagulant Activity. Int J Mol Sci. 2023;24(8):7163. doi:10.3390/ijms24087163). Some new inhibitors have already been applied to a model of Alzheimer's disease (Samantha S, Akhter F, Roy A, et al. Novel cyclophilin D inhibitor rescues mitochondrial and cognitive function in Alzheimer's disease. Brain. 2023. doi:10.1093/brain/awad432). Recently, the rational design of peptide inhibitors of CypD has also been described (Li Y, Liu T, Lai X, et al. Rational design of peptide inhibitors of cyclophilin D as a potential treatment for acute pancreatitis. Medicine (Baltimore). 2023; 102(48): e36188.doi: 10.1097 /MD.0000000000036188).
In addition, new inhibitors that do not target CypD have been described (Antonucci S, Di Sante M, Sileikyte J, et al. A new class of cardioprotective small molecule PTP inhibitors. Pharmacol Res. 2020;151:104548. doi:10.1016/j.phrs.2019.104548).
Author Response
We thank the reviewer for valuable comments.
Critics:
- We add more information about the physiological regulation of mPTP by mitochondrial membrane potential, redox state, pH, etc. to the revised version of the manuscript
- We thank the reviewer for pointing new non-immunosuppressive mPTP inhibitors out. We added information about it into the text and add some more references suggested by the reviewer
Reviewer 3 Report
Comments and Suggestions for Authors
The authors of the review paper “Mitochondrial permeability transition, cell death and neurodegeneration” have reviewed the involvement of mitochondrial dysfunction in the pathogenesis of the neurodegenerative disorders, with the emphasis on mitochondrial permeability transition (mPTP). They have also shortly discussed perspectives and difficulties in the development of the mPTP inhibitors for treatment of neurodegenerative disorders. However, more guidance from the authors throughout the paper is needed, with their stands and comments about up-to-date findings in the topic.
Comments:
1. The Abstract is very short, and some perspectives and the author stand for future use of the mPTP inhibitors in therapeutics of neurodegenerative disorders should be shortly stated.
2. In the introduction, the authors should mention recent statistics briefly, about incidence and prevalence of neurodegenerative diseases. Also, please state in the introduction most important genes/proteins associated with mitochondria (Mt) in neurodegenerative diseases.
3. List of abbreviations will be very helpful for the readers.
4. Two tables should be added to the review paper, one with most important literature in this topic and another that would compare similar and distinctive mitochondrial changes in different neurological conditions/disorders.
5. Line 488, ferutinin action should be explained exactly, as it is a selective estrogen receptor modulator, not an estrogen.
6. Conclusions of each section should be expanded and include short statement, i.e. author opinion of current knowledge on the topic and suggestions for future research.
7. English is relatively good in the paper, but some adjustments are needed.
Comments on the Quality of English LanguageEnglish is relatively good in the paper, but some adjustments are needed.
Author Response
We thank the reviewer for valuable comments. We address these comments below
- The Abstract is very short, and some perspectives and the author stand for future use of the mPTP inhibitors in therapeutics of neurodegenerative disorders should be shortly stated.We have extended the abstract but we do not want to excide the limits
2. In the introduction, the authors should mention recent statistics briefly, about incidence and prevalence of neurodegenerative diseases. Also, please state in the introduction most important genes/proteins associated with mitochondria (Mt) in neurodegenerative diseases.
We thank the reviewer for comment - we added this information to the sections related to diseases
3. List of abbreviations will be very helpful for the readers.
We added the list of abbreviations in the revised version of the manuscript
4. Two tables should be added to the review paper, one with most important literature in this topic and another that would compare similar and distinctive mitochondrial changes in different neurological conditions/disorders.
We thank the reviewer for suggestion. However, we have presented it in the form of figures
5. Line 488, ferutinin action should be explained exactly, as it is a selective estrogen receptor modulator, not an estrogen.
We thank the reviewer for pointing it out. It was corrected
6. Conclusions of each section should be expanded and include short statement, i.e. author opinion of current knowledge on the topic and suggestions for future research.
We added conclusions in the end of each section
7. English is relatively good in the paper, but some adjustments are needed.
It was corrected
Round 2
Reviewer 3 Report
Comments and Suggestions for Authors
Authors are answered all my concerns and corrected the paper accordingly. Although I think the readers would appreciate to have Tables with up-to-date literature, I do not insist on this request. It is true that the Figures cover part of it!
Comments on the Quality of English LanguageMinor editing of English language is required!